# Stratifying Disease Severity in Pediatric COVID-19: A Correlative Study of Serum Biomarkers and Lung Ultrasound—A Retrospective Observational Dual-Center Study

**DOI:** 10.3390/diagnostics14040440

**Published:** 2024-02-17

**Authors:** Emil Robert Stoicescu, Roxana Iacob, Adrian Cosmin Ilie, Emil Radu Iacob, Septimiu Radu Susa, Laura Andreea Ghenciu, Amalia Constantinescu, Daiana Marina Cocolea, Andreea Ciornei-Hoffman, Cristian Oancea, Diana Luminita Manolescu

**Affiliations:** 1Department of Radiology and Medical Imaging, ‘Victor Babes’ University of Medicine and Pharmacy Timisoara, Eftimie Murgu Square No. 2, 300041 Timisoara, Romania; stoicescu.emil@umft.ro (E.R.S.); dmanolescu@umft.ro (D.L.M.); 2Research Center for Pharmaco-Toxicological Evaluations, ‘Victor Babes’ University of Medicine and Pharmacy Timisoara, Eftimie Murgu Square No. 2, 300041 Timisoara, Romania; 3Faculty of Mechanics, Field of Applied Engineering Sciences, Specialization Statistical Methods and Techniques in Health and Clinical Research, ‘Politehnica’ University Timisoara, Mihai Viteazul Boulevard No. 1, 300222 Timisoara, Romania; 4Department of Anatomy and Embriology, ‘Victor Babes’ University of Medicine and Pharmacy Timisoara, 300041 Timișoara, Romania; 5IOSUD/Ph.D. School, ‘Victor Babes’ University of Medicine and Pharmacy Timisoara, Eftimie Murgu Square No. 2, 300041 Timisoara, Romania; septimiu.susa@umft.ro (S.R.S.); amalia.constantinescu@umft.ro (A.C.); daiana.cocolea@umft.ro (D.M.C.); 6Department III Functional Sciences, Division of Public Health and Management, “Victor Babes” University of Medicine and Pharmacy, 300041 Timisoara, Romania; ilie.adrian@umft.ro; 7Department of Pediatric Surgery, ‘Victor Babes’ University of Medicine and Pharmacy, Eftimie Murgu Square 2, 300041 Timisoara, Romania; radueiacob@umft.ro; 8Department of Functional Sciences, ‘Victor Babes’ University of Medicine and Pharmacy Timisoara, Eftimie Murgu Square No. 2, 300041 Timisoara, Romania; bolintineanu.laura@umft.ro; 9Department of Anatomy and Embryology, Morphological Sciences, ‘Iuliu Hatieganu’ University of Medicine and Pharmacy, 400349 Cluj-Napoca, Romania; andreea.hoffman@umfcluj.ro; 10Department of Radiology and Medical Imaging, County Clinical Emergency Hospital, 400006 Cluj-Napoca, Romania; 11Center for Research and Innovation in Precision Medicine of Respiratory Diseases (CRIPMRD), ‘Victor Babeș’ University of Medicine and Pharmacy, 300041 Timișoara, Romania; oancea@umft.ro; 12Department of Pulmonology, ‘Victor Babes’ University of Medicine and Pharmacy, 300041 Timisoara, Romania

**Keywords:** lung disease, thoracic ultrasound, lung ultrasound, neonates, infants, COVID-19, SARS-CoV-2, inflammatory markers, biomarkers, multisystem inflammatory syndrome

## Abstract

The COVID-19 pandemic, caused by SARS-CoV-2, has manifested distinct impacts on infants and children. This study delves into the intricate connection between lung ultrasound (LUS) findings and serum biomarkers in neonates and infants with COVID-19. Exploring factors contributing to the mild symptoms in this demographic, including immune responses and pre-existing immunity, the study spans 3 years and 9 months, involving 42 patients. Respiratory and gastrointestinal symptoms predominate, and LUS emerges as a vital, non-irradiating tool for evaluating pulmonary abnormalities. Serum biomarkers like CRP, procalcitonin, and cytokines provide key insights into the pathophysiology. Correlations reveal nuanced links between LUS score and clinical parameters, unveiling associations with hospitalization duration (rho = 0.49), oxygen saturation (rho = −0.88), and inflammatory markers, like ferritin (rho = 0.62), LDH (rho = 0.73), and D-dimer (rho = 0.73) with significance level (*p* < 0.05). The absence of large consolidations in LUS suggests unique pulmonary characteristics. The novelty of these findings lies in the comprehensive integration of LUS with serum biomarkers to assess and monitor the severity of lung involvement in neonates and infants affected by SARS-CoV-2. This approach offers valuable insights into disease severity, biomarker levels, the duration of hospitalization, and oxygen saturation, providing a multifaceted understanding of COVID-19’s impact on this vulnerable population.

## 1. Introduction

The COVID-19 pandemic, caused by the SARS-CoV-2 virus, has had a significant impact on global health, with various age groups being affected differently [1,2]. While adults have been more susceptible to severe forms of the disease, infants and children have generally exhibited milder symptoms [3,4]. Still, comprehending the association between serum biomarkers and lung ultrasonography findings in this susceptible population is crucial.

Children and neonates are less affected by COVID-19 due to several factors. One potential explanation is that children have a stronger innate immune response and a higher proportion of total lymphocytes, T cells, B cells, and natural killer cells, which helps them fight the virus. Additionally, children have a lower prevalence of co-morbidities that have been associated with severe disease in adults. Pre-existing immunity and cross-reacting antibodies for common circulating coronaviruses may also play a protective role in children. Furthermore, children are often infected by a second or third generation of the virus, which has been described as having decreased in pathogenicity. Moreover, the increased presence of viruses and bacteria in the mucosal lining of children may restrict the establishment and proliferation of SARS-CoV-2 due to microbial interactions and competition [3,5,6]. Another crucial factor is that children have demonstrated a lower expression of the angiotensin-converting enzyme II (ACE2) receptor, as indicated by several studies. The variation in the severity of COVID-19 between children and adults can be partially explained by the difference in the amount of ACE2, as well as TMPRSS2 expression in the airway tissues [7,8].

Previous studies have shown that neonates can acquire the infection through postnatal exposure, leading to a higher risk of infection within community or healthcare settings. Diagnosis methods commonly used in newborns include nasopharyngeal and rectal exudate samples, followed by RT-PCR tests. However, false negative results can occur during the incubation period of the virus [9,10,11]. Lung ultrasound (LUS) has emerged as a non-irradiating and repeatable imaging method for evaluating lung changes associated with respiratory pathologies [12,13,14,15]. It has the capacity to offer crucial insights into the pulmonary abnormalities reported in newborns and children with minimal mild signs of COVID-19 [13,16]. Its capacity to provide real-time, high-resolution images of the lungs has revolutionized respiratory diagnostics, enabling clinicians to visualize subtle structural abnormalities, monitor disease progression, and guide procedures with remarkable precision [17,18,19,20]. The operator-dependency of this evaluation has the potential to be significantly impacted in the future due to advancements in robotic ultrasound systems (RUSSs). These systems can be classified as either teleoperated or autonomous, and the importance of machine learning and artificial intelligence in facilitating smart image acquisition is highlighted [21,22]. Another game-changing aspect is the fact that in the upcoming five years, researchers are expected to concentrate on the development of five novel materials aimed at enhancing transmit–receive efficiencies, with a particular focus on creating transparent and flexible thin films for ultrasound acquisition [23].

Simultaneously, the examination of serum biomarkers, which are biological molecules that provide information about cellular processes, inflammation, and organ function, has become an additional aspect in understanding the complexity of the condition of the lungs [24,25,26]. These biomarkers, be it C-reactive protein (CRP), procalcitonin, or various cytokines and growth factors, serve as silent sentinels, providing invaluable cues about the underlying pathophysiology and disease progression [6,25,26]. By correlating LUS findings with serum biomarkers, such as LDH, D-dimer, and IL-6, a better understanding of disease severity and progression can be achieved [26,27,28].

This article aims to investigate the correlation between serum biomarkers and lung ultrasound findings in neonates and infants with COVID-19. By analyzing the LUS score and its relationship with inflammatory markers and clinical symptoms, we can potentially develop a severity score for the better evaluation of patients. Additionally, the study will explore the association between LUS score and oxygen saturation levels, as well as the impact of these findings on the duration of convalescence and hospitalization. Understanding the correlation between serum biomarkers and lung ultrasound in neonates and infants can contribute to the development of effective diagnostic and monitoring strategies for this vulnerable population, minimizing the utilization of radiation for diagnostic purposes.

By identifying specific biomarkers (leukocytes, lymphocytes, neutrophiles, ALT, AST, procalcitonin, CRP, ferritin, LDH, IL-6, d-dimer level) and ultrasound findings (interstitial, alveolar edema, or subpleural consolidation) associated with disease severity, healthcare providers can make informed decisions regarding the treatment and management of COVID-19 in infants and young children [27,29]. In the intricate landscape of respiratory medicine, the quest for more precise and comprehensive diagnostic tools has led to a promising union between cutting-edge imaging technology and the microscopic clues harbored within our bloodstream. This amalgamation manifests in the symbiotic relationship between lung ultrasound imaging and the intricate world of serum biomarkers, offering a transformative approach to understanding and managing lung diseases.

## 2. Materials and Methods

### 2.1. Study Design

This study utilized a retrospective observational design to examine the association between lung ultrasonography findings and serum biomarkers in patients with SARS-CoV-2 infection. The study was performed over a period of 3 years and 9 months (February 2020–November 2023) at the Neonatology and Neonatal Intensive Care Unit (NICU) at ‘Pius Brinzeu’ Emergency County Hospital and the Clinic of Infectious Diseases II and the Intensive Care Unit at ‘Dr. Victor Babes’ Clinical Hospital of Infectious Diseases and Pneumophthisiology in Timisoara, after clearance from the Ethics Committee and the obtention of informed consent from all participants.

### 2.2. Participant Selection

Participants were recruited in a sequential manner from the Neonatology and Neonatal Intensive Care Unit (NICU) and the Clinic of Infectious Diseases II and the Intensive Care Unit according to predetermined criteria for participation. These criteria included a specific age range (under one year), proof of SARS-CoV-2 infection, and willingness to undergo lung ultrasound imaging and blood sample collection.

Excluded from the study were:Hospitalized neonates and infants diagnosed with SARS-CoV-2 infection for a period shorter than three days;Neonates and infants with pre-existing chronic lung conditions like bronchopulmonary dysplasia, cystic fibrosis, immunodeficiency, and comparable disorders;Neonates and infants lacking parental or legal guardian consent.

### 2.3. Clinical and Laboratory Evaluations

Clinical and laboratory evaluations were conducted upon enrollment, which involved thorough assessments of participants’ medical history, physical condition, and pertinent diagnostic testing. Simultaneously, blood samples were taken using normal venipuncture procedures. The study involved analyzing certain serum biomarkers of interest, including hemoglobin level (g/dL), leukocyte count (×10^9^/L), lymphocyte count (×10^9^/L), neutrophile count (×10^9^/L), thrombocyte count (×10^9^/L), ALT level (U/L), AST level (U/L), total bilirubin (mg/dL), procalcitonin level (ng/mL), CRP level (mg/L), ferritin level (µg/L), LDH level (U/L), IL-6 level (pg/mL), and D-dimer level (mg/L), using laboratory techniques such as enzyme-linked immunosorbent assay (ELISA) and immunoassays. Stringent quality control methods were enacted to guarantee the precision and consistency of biomarker measurements.

### 2.4. Lung Ultrasound Examination

Lung ultrasound examinations were performed by skilled and certified radiologists with more than ten years’ experience, utilizing a specific ultrasound machine, settings, and probes. Regarding these, we used the portable machine General Electric Vivid IQ that is furnished with a linear probe (9L-RS [2.4–10.0 MHz]) and a convex probe (4C-RS [1.5–5.0 MHz]). Additionally, the ultrasound system Philips EPIQ 5 that is equipped with the L12-5 linear array probe ([12–5 MHz]) was used. All the examinations were performed using the lung presetting protocol provided by the manufacturer and improved according to the needs of the patient. The focus was directed towards the pleural line, with the goal of achieving clear visualization of the hyperechoic line. The exams were concentrated on specified lung regions, adhering to established protocols.

### 2.5. Lung Ultrasound Score and Protocol

Every infant or newborn admitted to the hospital underwent a thorough lung evaluation using a 12-area scoring system. This scoring system was similar to the one outlined by Mongodi et al. for COVID-19-related pneumonia in neonates (referred to as the Lung Ultrasound Score) that covered six areas on each side of the chest (two anterior, two lateral, and two posterior) delineated by the nipple line [30]. Within each explored area, a scoring system ranging from 0 to 3 points was applied, based on the observation of artifacts and the presence or absence of subpleural consolidation:LUS score = 0 was assigned for a normal or physiological pattern displaying A-lines, along with one or two B-lines per intercostal space;LUS score = 1 indicated the observation of more than two B-lines (referred to as sparse B-lines) per intercostal space, accompanied by pleural abnormalities, such as irregularities or thickening;LUS score = 2 was allocated for the presence of coalescent or merging B-lines, a ‘white-lung’ appearance, or small peripheral consolidations smaller than 1 cm;LUS score = 3 was given for substantial peripheral consolidations wider than 1 cm, regardless of the presence of air bronchograms.

An illustrated explanation of the LUS score is provided in Table 1.

This LUS scoring system enabled a detailed and nuanced assessment of lung conditions, providing a comprehensive summary of each patient’s lung ultrasound findings.

### 2.6. Data Collection and Analysis

Clinical data, ultrasound findings, and biomarker measurements were meticulously documented in a secure computerized database using Microsoft Excel. MedCalc^®^ Statistical Software version 22.017 (MedCalc Software Ltd., Ostend, Belgium; https://www.medcalc.org; accessed on 12 January 2024) was utilized to conduct statistical analyses in order to investigate the link between lung ultrasonography observations, LUS score and serum biomarker levels.

The study employed correlation coefficients, such as Spearman’s rank correlation (rho), to evaluate the magnitude and direction of the relationships between ultrasound findings and biomarker concentrations.

### 2.7. Ethical Considerations

The study followed the criteria stated in the Declaration of Helsinki. The study rigorously upheld patient anonymity, ensuring that data were de-identified for analysis and publishing.

The research was carried out in university hospitals, and it was necessary to secure the patient’s agreement in every instance. Consent for infants and newborns was acquired from their parents, guardians, or the person responsible for their care. After giving the caregivers essential information, consent was obtained from the caregivers of all the individuals who participated in the study. The patient’s guardians have given fully informed written consent for the publication of this study. 

## 3. Results

### 3.1. Baseline Characteristics

Out of a cohort of 42 patients, including newborns and infants, 24 were male, accounting for 57.14% of the total.

The median value of total PCR tests performed was 2, with the IQR of [1;3]. The lowest value recorded was 1, and the highest value recorded was 7. The median value of positive PCR tests conducted was 2, with an interquartile range (IQR) of [1;2]. The minimum value recorded was 1, while the maximum value recorded was 4.

The median figure for the duration of hospitalization was 5.50 days, with an IQR of 4 to 9 days. The maximum duration of hospitalization was 28 days, but the minimum was a mere two days.

### 3.2. Signs and Symptoms

Table 2 displays the most reliable indicators and symptoms examined for newborns and infants.

### 3.3. Lung Ultrasound Score and Ultrasound Abnormalities

Table 3 presents the lung ultrasonography results and their incidence. The identified observations include sparse B-lines, confluent B-lines, pleural anomalies, subpleural consolidation measuring less than 1 cm, significant consolidation, and pleural effusion. Furthermore, Table 3 displays the overall LUS score and the demarcation of areas of interest utilizing the lung ultrasound approach.

### 3.4. Lung Ultrasound Score and Correlation with Inflammatory Markers

Table 4 presents the correlation between LUS score and principal parameters and biomarkers. The analyzed data include the days of hospitalization, weight, hemoglobin, leukocytes count, lymphocytes count, neutrophiles count, thrombocytes count, ALT level, AST level, total bilirubin level, procalcitonin level, CRP level, ferritin level, LDH level, IL-6 level, D-dimer level, and O2 saturation.

## 4. Discussion

The data analysis reveals that the cohort’s testing frequency is moderate, and there is significant variability in the number of positive outcomes. The duration of hospital admissions demonstrated notable variation, from 2 to 28 days, suggesting a spectrum of disease severity. These findings emphasize the importance of tailoring patient care to meet individual conditions. These insights are crucial for enhancing the allocation of resources and optimizing treatment processes in comparable populations [5,9,31].

A notable proportion of the cohort exhibited moderately influenced general conditions, indicative of a range of symptoms affecting overall wellbeing. Psychomotor agitation and asthenic syndrome were prevalent, underscoring the complexity of clinical presentations in this age group. Respiratory symptoms, including fever, cough, and rhinorrhea, were common, reflecting the susceptibility of neonates and infants to respiratory manifestations during SARS-CoV-2 infection. Gastrointestinal manifestations, such as mild acute dehydration, episodes of diarrhea, vomiting, and loss of appetite, indicated a significant impact on the gastrointestinal tract. These manifestations bear resemblance to other articles with a proportional comparison of prevalence [32,33].

Notably, a small subset displayed signs of moderate acute dehydration, emphasizing the importance of monitoring and addressing hydration status [31]. Additionally, the presence of dyspnea highlighted potential respiratory distress in a subset of patients [34]. Other notable findings include lateral cervical lymph node involvement and oral candidiasis. This diversity in symptoms emphasizes the challenges in diagnosing and managing illnesses in neonates and infants, necessitating a nuanced approach to address the multifaceted nature of their clinical presentations.

The lung ultrasound findings reveal a comprehensive spectrum of pulmonary abnormalities. Sparse B-lines were universally observed in all patients, indicating a consistent ultrasonographic feature across the cohort [11,27,29,35]. Confluent B-lines, pleural abnormalities, and subpleural consolidations measuring less than 1 cm were prevalent in varying percentages, suggesting a range of lung involvement [16,35].

Notably, large consolidations measuring more than 1 cm were absent in all cases, possibly indicating a specific characteristic of the pulmonary pathology in this cohort. Ibarra-Ríos et al. found that significant consolidation had a prevalence of 37%, with the threshold for defining it set arbitrarily at 5 mm [36]. Pleural effusion, although relatively infrequent, was noted in a small subset of patients [27,29,36]. These findings contribute to a nuanced understanding of the pulmonary manifestations, serving as essential indicators for the overall Lung Ultrasound Score. The absence of large consolidations may offer insights into the nature of lung involvement in this specific patient population. The comprehensive ultrasonography evaluation is crucial for providing information that guides therapeutic decision making and customizing therapies according to the identified pulmonary abnormalities [12,13,19,36].

The lung ultrasound findings across distinct anatomical areas provide a comprehensive understanding of the pulmonary involvement in the studied population. The total LUS score across all areas amounted to 337, indicating a cumulative assessment of the severity of lung abnormalities. The distribution of scores across specific lung regions illustrates varied degrees of involvement. Notably, the posterior inferior regions of both the left (L6) and right (R6) lungs had the highest individual scores, suggesting a predilection for abnormalities in these areas. This regional analysis enables a more detailed assessment of the lung pathology, yielding conclusions that are consistent with the results of other articles [13,16,17]. The percentages assigned to each area within the total LUS score highlight the proportional contribution of specific lung regions to the overall severity score. The comprehensive anatomical insights obtained from lung ultrasound play a crucial role in customizing treatment regimens and monitoring the evolution of diseases in patients, hence improving the accuracy of therapeutic management [36,37,38].

The Spearman’s rank correlation coefficients between the LUS score and a range of clinical parameters shed light on the intricate relationships between lung involvement and systemic health indicators in the studied cohort. A positive correlation was notably observed with the duration of hospitalization (rho = 0.49), indicating that as the LUS score increases, patients tend to have a more prolonged hospital stay. A robust negative correlation between LUS score and weight (rho = -0.72) suggests that a higher lung severity is associated with lower body weight, emphasizing the systemic impact of respiratory distress. Among hematological parameters, positive correlations with leukocytes (rho = 0.48), neutrophils (rho = 0.38), and total bilirubin (rho = 0.49) imply a link between lung involvement and heightened immune response and liver function. The substantial positive correlations with procalcitonin (rho = 0.35), CRP (rho = 0.34), ferritin (rho = 0.62), LDH (rho = 0.73), IL-6 (rho = 0.46), and D-dimer (rho = 0.73) underscore the inflammatory and prothrombotic aspects associated with an increased LUS score. These findings suggest that as the severity of lung abnormalities rises, so does the overall inflammatory burden and coagulation activation [26,39,40]. Importantly, the strong negative correlations between the LUS score and oxygen saturation (rho = -0.88) emphasize the critical impact of lung involvement on respiratory function. The results collectively highlight the utility of the LUS score as a comprehensive tool, not only reflecting the severity of lung pathology but also providing insights into the systemic implications and potential clinical outcomes in patients [31,36].

The moderate positive correlation with the duration of hospitalization implies that the LUS score may serve as an early indicator of the expected length of the hospital stay. The predictive nature of this element is beneficial for both physicians and patients, as it assists in the allocation of resources and establishes reasonable expectations for the progression of the condition [27,29,41].

The strong negative correlation between LUS score and weight not only suggests a link between the severity of lung involvement and nutritional status but also implies that respiratory distress may contribute to weight loss in neonates. Furthermore, newborns exhibited a higher lung ultrasound score in comparison to infants, as determined by their weight [27,29,31,36,38].

The presence of positive correlations between biomarkers such as LDH and D-dimer indicates that the LUS score not only reflects morphological changes visible via ultrasound but also corresponds to biochemical indicators associated with tissue injury and coagulation activation. This offers a comprehensive perspective on the severity of the disease and its overall effects on the body [26,27,29]. The robust negative correlation with oxygen saturation is particularly noteworthy. It indicates that the LUS score can effectively capture the decline in respiratory function, aligning with the well-established understanding that as lung involvement worsens, oxygen saturation tends to decrease. This could be crucial in identifying patients at higher risk of hypoxemia. Clinically, this underscores the systemic impact of severe lung conditions on overall health. Clinicians can use LUS score measurements to anticipate the risk of hypoxemia, enabling prompt interventions and the close monitoring of patients at higher risk. Another crucial aspect that warrants discussion is the heightened association between LUS score and O2 saturation, surpassing the findings of previous research that examined neonates and infants as different populations [27,29,36,42].

The positive correlations with inflammatory markers (CRP, Ferritin, IL-6, Procalcitonin) highlight the LUS score ‘s utility in monitoring the inflammatory status of patients. This information can guide clinicians in adjusting treatment strategies based on the evolving inflammatory response, providing a dynamic approach to patient management [40,43,44].

The crucial factor for achieving improved patient management and medical care is in the collaborative efforts of multiple disciplines, as indicated by the data on the connections between LUS and biomarkers. Radiologists and neonatologists collaborate closely in lung ultrasound examinations, particularly in neonatal care settings, to leverage their respective expertise for comprehensive patient evaluation and management. Radiologists, with their specialized training in medical imaging interpretation, provide detailed analysis of ultrasound images, identifying lung abnormalities and offering insights into their nature and extent. Neonatologists, on the other hand, contribute valuable clinical context, integrating imaging findings with the infant’s medical history, symptoms, and other diagnostic tests to formulate a holistic assessment and treatment plan. This collaborative approach ensures that diagnostic decisions are made in real time, optimizing patient care and outcomes in the neonatal intensive care unit (NICU) by combining imaging expertise with clinical knowledge for tailored and effective interventions [45,46].

### 4.1. Limitations

Sample size: while our study boasts the largest sample size in this comprehensive investigation of infants and neonates, its size may still constrain the generalizability of findings. A more extensive participant pool could offer a more nuanced understanding of observed disparities.

Limited data on long-term outcomes: the study’s focus on immediate clinical presentations might lead to the oversight of potential long-term effects or outcomes in infants and neonates post infection.

High sensitivity, a little lower specificity for LUS: we have to admit that the contrast between pulmonary edema and interstitial patterns caused by other factors is evident; however, the distinction between viral pneumonia and interstitial patterns caused by various factors may not be as apparent [47].

### 4.2. Further Directions

Expanded comparative studies: enlarging the scale of studies to include more diverse cohorts of infants and neonates could provide a more comprehensive understanding of the nuances in lung involvement and validate the observed trends.

Longitudinal LUS investigations: examining the evolution of LUS findings over time in infants and neonates with SARS-CoV-2 could offer insights into the progression or resolution of lung abnormalities.

Comprehensive LUS protocols in accordance with serum biomarkers: developing comprehensive LUS protocols that encompass a wider array of lung pathology and standardizing scoring systems could enhance the accuracy and reproducibility of assessments.

Focusing on these aspects in future studies could refine the role of LUS in assessing lung involvement in infants and neonates affected by SARS-CoV-2 and other respiratory diseases, enhancing its applicability, and contributing to more robust clinical practices for managing these vulnerable populations. Furthermore, the advancement of handheld ultrasound devices has the potential to enhance the collection of data and images, as well as facilitating the creation of large-scale databases through improved accessibility [48].

## 5. Conclusions

In summary, the findings underscore the multi-dimensional utility of the LUS score as a tool for assessing and monitoring the severity of lung involvement. This study highlights the valuable integration of lung ultrasound and serum biomarkers (LDH, D-dimer, ferritin) for a comprehensive assessment of SARS-CoV-2’s impact on neonates and infants. The diverse clinical presentations and lung abnormalities emphasize the complex nature of COVID-19 in this population. Key correlations between LUS score and clinical parameters offer insights into the disease severity, duration of hospitalization, and oxygen saturation.

## Figures and Tables

**Figure 1 diagnostics-14-00440-f001:**
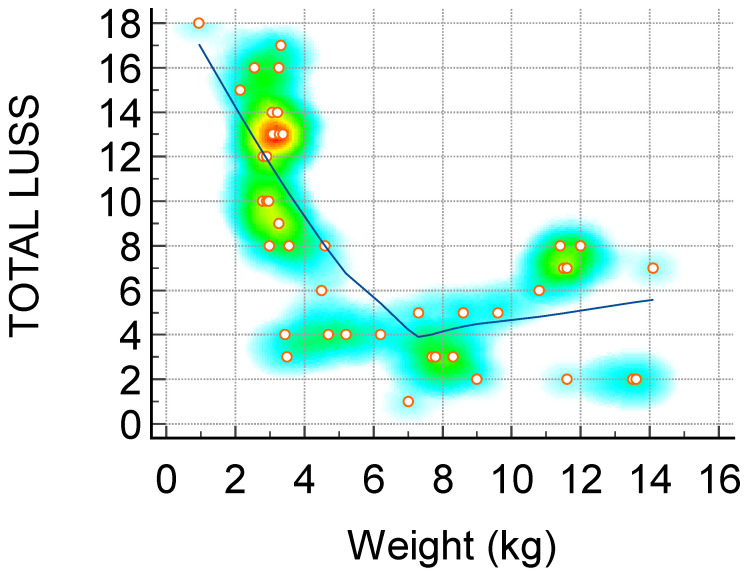
Scatter diagram with heat map of correlation between LUS score and weight (kg)—a negative linear correlation. The background color coding indicates density of points, suggesting clusters of observations. The red color indicates a high concentration of points, yellow indicates a moderate concentration of points, and blue indicates a low concentration of points.

**Figure 2 diagnostics-14-00440-f002:**
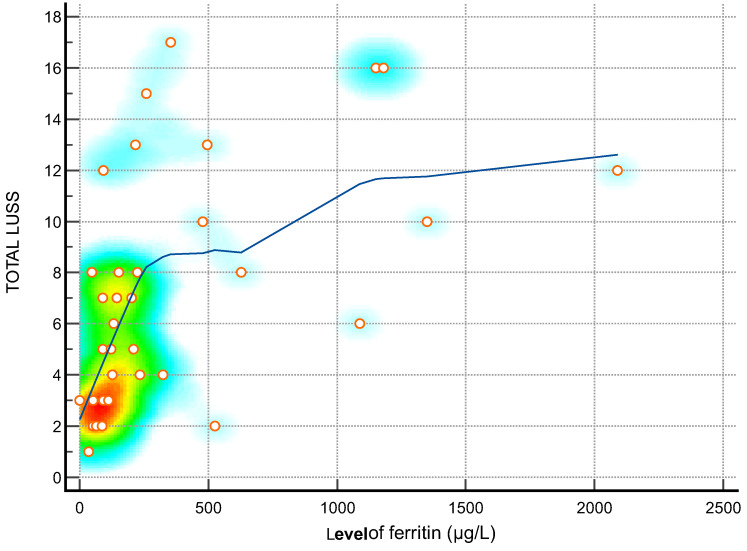
Scatter diagram with heat map of correlation between LUS score and level of ferritin—a positive linear correlation. The background color coding indicates density of points, suggesting clusters of observations. The red color indicates a high concentration of points, yellow indicates a moderate concentration of points, and blue indicates a low concentration of points.

**Figure 3 diagnostics-14-00440-f003:**
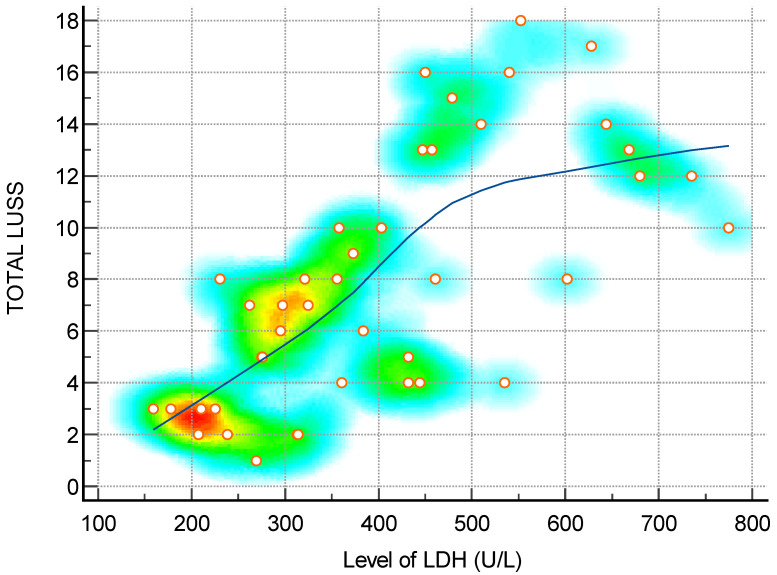
Scatter diagram with heat map of correlation between LUS score and level of LDH—a positive linear correlation. The background color coding indicates density of points, suggesting clusters of observations. The red color indicates a high concentration of points, yellow indicates a moderate concentration of points, and blue indicates a low concentration of points.

**Figure 4 diagnostics-14-00440-f004:**
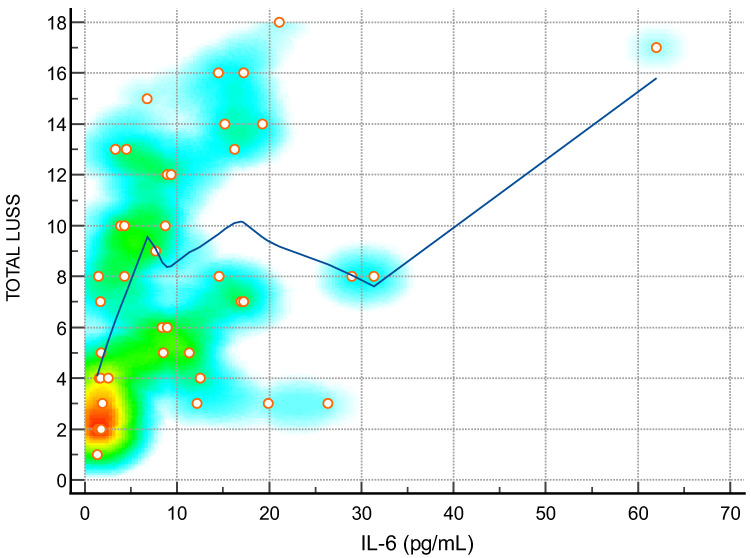
Scatter diagram with heat map of correlation between LUS score and level of IL-6—a positive linear correlation. The background color coding indicates density of points, suggesting clusters of observations. The red color indicates a high concentration of points, yellow indicates a moderate concentration of points, and blue indicates a low concentration of points.

**Figure 5 diagnostics-14-00440-f005:**
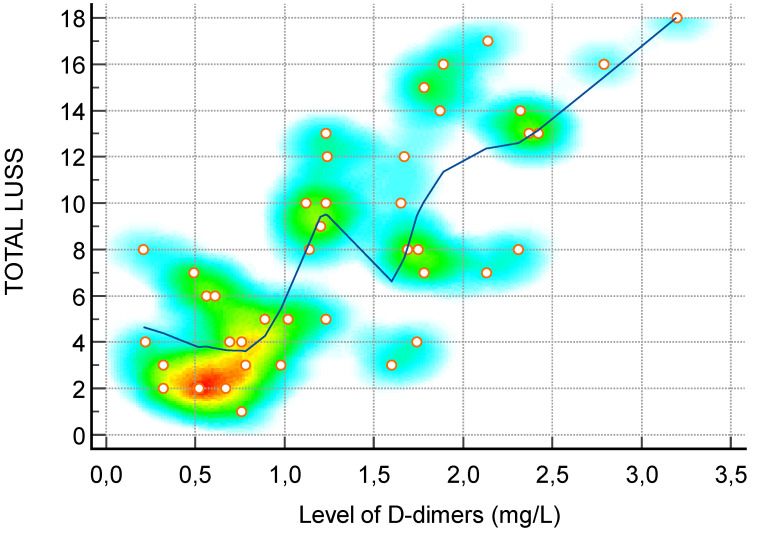
Scatter diagram with heat map of correlation between LUS score and level of D-dimers—a positive linear correlation. The background color coding indicates density of points, suggesting clusters of observations. The red color indicates a high concentration of points, yellow indicates a moderate concentration of points, and blue indicates a low concentration of points.

**Figure 6 diagnostics-14-00440-f006:**
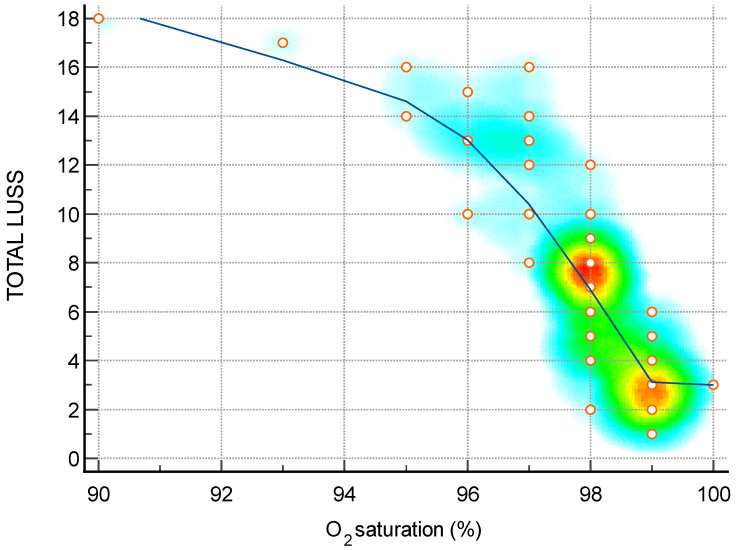
Scatter diagram with heat map of correlation between LUS score and O_2_ saturation—a negative linear correlation. The background color coding indicates density of points, suggesting clusters of observations. The red color indicates a high concentration of points, yellow indicates a moderate concentration of points, and blue indicates a low concentration of points.

**Table 1 diagnostics-14-00440-t001:** An illustrated explanation of the LUS score.

LUS Score	0 Points	1 Point	2 Points	3 Points
Image	** 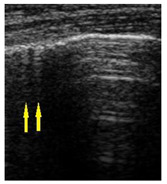 **	** 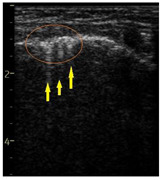 **	** 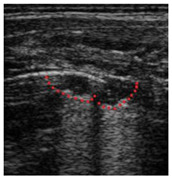 **	** 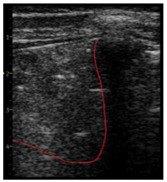 **
Description of image	Normal or physiological pattern displaying A-lines (right part), along with two sparse B-lines (yellow arrows) per intercostal space	Three sparse B-lines (yellow arrows) per intercostal space, accompanied by pleural abnormalities, such as irregularities or thickening (orange circle)	Small peripheral consolidations smaller than 1 cm (red dotted area), a small area with ‘white-lung’ appearance, adjacent coalescent or merging B-lines	Substantial peripheral consolidations (marked with red) wider than 1 cm with the presence of air bronchograms (hyperechoic areas inside); the image is from an infant with bacterial pneumonia not included in this study

**Table 2 diagnostics-14-00440-t002:** The signs and symptoms in infants and children with SARS-CoV-2 infection presented as the number of patients and percentage (%)**.**

Signs and Symptoms in Neonates and Infants	*n* = 42 (Percentage %)
Moderately influenced general condition	27 (64.28)
Slightly influenced general condition	15 (35.71)
Psychomotor agitation	16 (38.09)
Asthenic syndrome	14 (33.33)
Fever (≥37.5 °C)	26 (61.90)
Cough	17 (40.47)
Rhinorrhea	18 (42.85)
Mild acute dehydration syndrome (<5% of weight)	24 (57.14)
Moderate acute dehydration syndrome (5–10% of weight)	2 (4.76)
Episodes of diarrhea	9 (21.42)
Vomiting	6 (14.28)
Loss of appetite	23 (54.76)
Lateral cervical lymph nodes	6 (14.28)
Dyspnea	4 (9.52)
Oral candidiasis	12 (28.57)

**Table 3 diagnostics-14-00440-t003:** The incidence of lung ultrasound findings. Areas of interest in lung ultrasound examination.

LUS Findings	*n* = 42 (Percentage %)
Sparse B-lines	42 (100)
Confluent B-lines	18 (42.85)
Pleural abnormalities	23 (54.76)
Subpleural consolidation < 1 cm	10 (23.80)
Large consolidation < 1 cm	0
Pleural effusion	2 (4.76)
Areas of interest in lung ultrasound	Total LUS score (percentage % from total LUS score)
All areas	337
L1—left anterior superior	19 (5.63)
L2—left anterior inferior	22 (6.52)
L3—left lateral superior	25 (7.41)
L4—left lateral inferior	25 (7.41)
L5—left posterior superior	31 (9.19)
L6—left posterior inferior	38 (11.27)
R1—right anterior superior	27 (8.01)
R2—right anterior inferior	26 (7.71)
R3—right lateral superior	20 (5.93)
R4—right lateral inferior	24 (7.12)
R5—right posterior superior	38 (11.27)
R6—right posterior superior	42 (12.46)

**Table 4 diagnostics-14-00440-t004:** The correlation between LUSs and principals parameters and biomarkers.

Correlation between LUS Score and the below Variables	Spearman’s Coefficient of Rank Correlation (rho)	Rho 95% Confidence Interval	Significance Level *p* Value
Days of hospitalization	0.49	0.21 to 0.69	0.0010
Weight (kg)—Figure 1	−0.72	−0.84 to −0.53	<0.0001
Hemoglobin (g/dL)	0.30	−0.00 to 0.55	0.0520
Leukocytes (×10^9/^L)	0.48	0.21 to 0.69	0.0010
Lymphocytes (×10^9/^L)	0.32	0.02 to 0.57	0.0348
Neutrophiles (×10^9/^L)	0.38	0.08 to 0.61	0.0127
Thrombocytes (×10^9^/L)	−0.22	−0.49 to 0.08	0.1568
ALT(U/L)	0.11	−0.19 to 0.40	0.4578
AST (U/L)	0.30	0.00 to 0.56	0.0462
Total bilirubin (mg/dL)	0.49	0.18 to 0.71	0.0036
Procalcitonin (ng/mL)	0.35	0.00 to 0.62	0.0487
CRP (mg/L)	0.34	0.04 to 0.58	0.0267
Ferritin (µg/L)—Figure 2	0.62	0.36 to 0.79	0.0001
LDH (U/L)—Figure 3	0.73	0.56 to 0.85	<0.0001
IL-6 (pg/mL)—Figure 4	0.46	0.19 to 0.67	0.0017
D-dimer (mg/L)—Figure 5	0.73	0.55 to 0.85	<0.0001
O_2_ saturation (%)—Figure 6	−0.88	−0.93 to −0.79	<0.0001

## Data Availability

The data are encapsulated within the article. Further details can be obtained upon request from either the primary author or the corresponding author. The data are inaccessible to the public due to the patient privacy regulations governing clinical data.

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
