# Peer review of "Stratifying Disease Severity in Pediatric COVID-19: A Correlative Study of Serum Biomarkers and Lung Ultrasound—A Retrospective Observational Dual-Center Study"

_diagnostics, 2024, doi:10.3390/diagnostics14040440_

Round 1

Reviewer 1 Report

Comments and Suggestions for Authors

Suggested changes:

1) Please modify the abstract to convey the novelty of the manuscript only. 

2) Please increase the number of figures

3) Pg 3, line 102: "By identifying specific biomarkers and ultrasound findings associated with disease". Please elaborate on the biomarkers and the ultrasound findings here itself.

4) Authors need to describe the lung ultrasound score better in a separate paragraph

5) For all the figures/plots, it is important to elaborate in separate paragraphs as what is the reason of getting the observed trend?

6) It is encouraged to add recent development in general ultrasound domain:

(a) https://doi.org/10.1016/j.media.2023.102878

(b) https://doi.org/10.1038/s41378-023-00555-7

(c) 10.1109/TUFFC.2023.3327143

Author Response

Dear reviewer,

Thank you very much for these valuable comments.

We carefully reviewed the text and made necessary corrections to the English language.

We appreciate your ongoing support in enhancing general comprehension of the data presented in the manuscript.

We hope that the changes that have been implemented are to your satisfaction, thereby ensuring that the article fulfills the requisite criteria for publication consideration.

Sincerely,

Authors

Reviewer 2 Report

Comments and Suggestions for Authors

I read with attention the article entitled. 

 Stratifying Disease Severity in Pediatric COVID-19: A Correla-2 tive Study of Serum Biomarkers and Lung Ultrasound.

The article is well-written and well-presented, but the sample size is low. The authors correctly underline this aspect in the study limitation.

My suggestions:

1) Use the LUS score and not LUSS in the article

2) Please add the type of the study in the title 

2) Please better explain that you obtained for all parents the consent in a retrospective study

3) Please better explain the fact that radiologists and neonatology work together 

4) Please better explain the fact that children showed less ACE receptor and are less affected

5) Please add this in the study limitation "Lung Ultrasound and the COVID-19 "Pattern": Not All That Glitters Today Is Gold Tomorrow. J Ultrasound PMID: 32383793; PMCID: PMC7272952.

Best Regards

Comments on the Quality of English Language

Accettable

Author Response

(The authors gave the same response as above.)

Round 2

Reviewer 1 Report

Comments and Suggestions for Authors

The manuscript is in good condition now